# NK Cells and Other Cytotoxic Innate Lymphocytes in Colorectal Cancer Progression and Metastasis

**DOI:** 10.3390/ijms23147859

**Published:** 2022-07-16

**Authors:** Cinzia Fionda, Gianluca Scarno, Helena Stabile, Rosa Molfetta, Chiara Di Censo, Angela Gismondi, Rossella Paolini, Silvano Sozzani, Angela Santoni, Giuseppe Sciumè

**Affiliations:** 1Department of Molecular Medicine, Sapienza University of Rome, Viale Regina Elena, 291, 00161 Rome, Italy; gianluca.scarno@uniroma1.it (G.S.); helena.stabile@uniroma1.it (H.S.); rosa.molfetta@uniroma1.it (R.M.); chiara.dicenso@uniroma1.it (C.D.C.); angela.gismondi@uniroma1.it (A.G.); rossella.paolini@uniroma1.it (R.P.); silvano.sozzani@uniroma1.it (S.S.); angela.santoni@uniroma1.it (A.S.); 2Istituto Pasteur Italia—Fondazione Cenci-Bolognetti, Viale Regina Elena, 291, 00161 Rome, Italy; 3IRCCS Neuromed, Pozzilli, 86077 Isernia, Italy

**Keywords:** colorectal cancer, inflammation, natural killer cells, innate lymphoid cells

## Abstract

Colorectal cancer (CRC) is one of the most common malignancies and leading causes of cancer-related deaths worldwide. Despite its complex pathogenesis and progression, CRC represents a well-fitting example of how the immune contexture can dictate the disease outcome. The presence of cytotoxic lymphocytes, both CD8^+^ T cells and natural killer (NK) cells, represents a relevant prognostic factor in CRC and is associated with a better overall survival. Together with NK cells, other innate lymphocytes, namely, innate lymphoid cells (ILCs), have been found both in biopsies of CRC patients and in murine models of intestinal cancer, playing both pro- and anti-tumor activities. In particular, several type 1 innate lymphoid cells (ILC1) with cytotoxic functions have been recently described, and evidence in mice shows a role for both NK cells and ILC1 in controlling CRC metastasis. In this review, we provide an overview of the features of NK cells and the expanding spectrum of innate lymphocytes with cytotoxic functions. We also comment on both the described and the potential roles these innate lymphocytes can play during the progression of intestinal cancer leading to metastasis. Finally, we discuss recent advances in the molecular mechanisms underlying the functional regulation of cytotoxic innate lymphocytes in CRC.

## 1. Introduction

Colorectal cancer (CRC) is the third most common malignancy and one of the leading causes of cancer-related deaths, responsible for almost 1 million annual deaths world-wide, according to the International Association of Cancer Registries [1]. The majority of CRC cases are associated with sporadic mutations linked to risk factors or lifestyle; 10–30% of the cases, instead, present family history, while less than 5% of patients show hereditary forms of the disease [2,3,4]. Diet, smoke, alcohol consumption, as well as chronic inflammation in patients with inflammatory bowel disease (IBD), all represent critical independent risk factors for CRC development [5,6,7,8,9]. CRC pathogenesis and progression are driven by distinct genetic features and events of genomic instability which lead to different CRC phenotypes characterized by chromosomal instability (CIN), hypermethylation of promoter CpG island sites (CpG island methylator phenotype, CIMP), and a high level of microsatellite instability (MSI-High) [3,10,11]. Despite these genetic features, CRC represents a well-fitting example of how the immune contexture can also dictate the outcome of tumor progression. Indeed, along with CRC classifications based on the features of cancer cells, including microsatellite instability and TNM staging, the WHO has introduced the “immunoscore” as a prognostic value for predicting disease-specific recurrence and survival [12,13,14,15,16]. Among the parameters, this immunoscore includes the density of tumor-infiltrating cytotoxic and memory T cells, which are associated with favorable prognosis [17]. Moreover, based on the Cancer Genome Atlas Consortium (TCGA) transcriptomic datasets, six immune subtypes of the tumor microenvironment (TME) have been identified, named wound healing, IFN-γ dominant, inflammatory, lymphocyte-depleted, immunologically quiet, and TGF-β dominant [18]. Besides TME and immune contexture, this classification also considers genetics.

By employing the same effector machinery of CD8^+^ cytotoxic T lymphocytes, Natural Killer (NK) cells represent the prototype of cytotoxic innate lymphocytes [19], and their frequency has been associated with a favorable outcome in CRC patients [20,21]. Together with NK cells, innate lymphoid cells (ILCs) have been also found both in tumor biopsies of CRC patients and in mouse models of CRC, playing both pro- and anti-tumor activities [22,23,24,25]. Added complexity is provided by the identification of ILCs expressing perforin and granzymes able to kill target cells, as well as by evidence in mice showing that NK cells and cytotoxic ILC can limit cancer growth [26,27,28,29,30,31,32,33,34,35]. In this review, we provide an overview of the features of NK cells and the expanding spectrum of innate lymphocytes with cytotoxic functions. We also discuss the potential roles of these cells in the progression and metastasis of intestinal cancer, as well as recent advances in the molecular mechanisms underlying NK cell functions in CRC.

## 2. NK Cells and the Expanding Family of Cytotoxic Innate Lymphocytes

NK cells remained the only known cell type belonging to the innate lymphoid branch until 2008, when several ILC populations able to elicit polarized effector phenotypes started to be discovered [33,36,37]. Paralleling the functions of CD8^+^ cytotoxic T cells and CD4^+^ helper T cells, innate lymphocytes are currently classified in five prototypical subsets: cytotoxic NK cells; three “helper-like” subsets, namely, ILC1, ILC2 and ILC3; and lymphoid tissue-inducer (LTi) cells, which represent a separate lineage involved in the generation of secondary lymphoid organs [38]. This nomenclature contemplates not only the functional properties, but also the distinct developmental trajectories of NK cells and ILC subsets and their different transcriptional requirements. NK cells and ILC1 were originally grouped together as type 1 ILCs because of their ability to produce Interferon (IFN)-γ as well as for their shared expression of the lineage-defining transcription factor (TF) T-bet and many surface makers, including natural cytotoxicity receptor 1 (NCR1, NKp46) and NK1.1 in C57Bl/6 mice [39,40,41,42]. This transcriptional similarity has made it difficult to specifically target ILC1 by using genetic tools in mice without affecting NK cells, except for the TF Hobit which is selectively required by liver ILC1 and not by NK cells or other known tissue-resident ILC1 [30,43,44,45]. However, unlike NK cells, ILC1 show limited ability to circulate in the peripheral blood (PB) at steady state and have been considered poor cytotoxic cells [4,41].

Advances in single-cell transcriptomic approaches have helped to refine the phenotypes and the identity of NK cells and other innate lymphocytes in humans and mice [46,47,48]. Human PB NK cells include several subsets corresponding to distinct differentiation stages and are divided, by convention, in two major subsets based on the expression of CD56, namely, CD56^bright^ and CD56^dim^ NK cells [49,50]. CD56^dim^ NK cells represent around 90% of total PB NK cells and are characterized by the high expression of CD16 and a higher potential to kill tumor cells, while CD56^bright^ NK cells express higher levels of CD94-NKG2A heterodimers and primarily produce large amounts of cytokines including IFN-γ and tumor necrosis factor (TNF)-α [51,52]. These NK cell subsets also differ for the expression of chemokine receptors and homing properties as well as for transcriptomic and epigenetic regulatory programs [47,53,54,55,56,57]. In mice, three main NK cell subsets have been identified according to the expression of CD27, a member of the TNF receptor superfamily, and the integrin chain CD11b; these cells also differ for functions and homing receptors [58,59,60,61].

NK cell activation is regulated by a dynamic balance between positive and negative signals from cell surface activating and inhibitory receptors which recognize ligands on potential target cells; their functions have been intensively reviewed and are only discussed briefly below (interested readers are referred to other outstanding reviews [62,63,64,65]). NK cells mainly receive inhibitory signals by the leukocyte antigen class I (HLA-I)-binding receptors, including killer cell Ig-like receptors (KIRs) in humans and Ly49 receptors in mice, and by CD94/NKG2A heterodimers [66,67]. KIR and Ly49 receptors bind HLA-I(A-C) and H-2K and H-2D, respectively, while CD94/NKG2A heterodimers recognize human HLA-E and mouse Qa1. These interactions transmit negative signals that limit NK cell function. This inhibition is required both to prevent NK cell responses against healthy cells as well as to ensure functional competence during development, via a process named education or licensing [68,69,70,71]. The frequent downregulation or absence of HLA-I molecules on virus-infected and neoplastic cells limit the delivery of inhibitory signals to NK cells. Moreover, during NK cell activation, other immune checkpoint receptors can be upregulated, as well as their ligands, on target cells. These include T-cell immunoglobulin and ITIM domain (TIGIT) and Programmed Cell Death Protein 1 (PD-1) and their ligands, CD155 and PDL-1/2 in virus-infected and tumor cells [66,72]. Activating signals, instead, are determined by a wide array of receptors, such as Natural Killer receptor group 2, member D (NKG2D), DNAX accessory molecule-1 (DNAM-1), natural cytotoxicity receptors (NCRs: NKp30, NKp44 and NKp46), CD94/NKG2C, and KIRs with activating intracellular domains [66,73]. A peculiarity of these receptors consists in their ability to interact with two or more ligands. For instance, NKG2D ligands include proteins major histocompatibility complex (MHC) class I-related chain A (MICA) and B (MICB) and UL16-binding proteins (ULBP1-6), while DNAM-1 ligands consist of CD155 and CD112. Tumor transformation or viral infection leads to the upregulation or neo-induction of ligands of different NK cell activating receptors. Engagement of these activating receptors by their cognate ligands promotes NK cell-mediated recognition and killing of target cells. Additionally, the low-affinity FcγRIIIA receptor (CD16) allows NK cells to exert antibody-dependent cellular cytotoxicity (ADCC) [74]. Notably, different levels of activation associated with distinct functional outcomes can be obtained by the concomitant triggering of two or more NK cell activating receptors [75]. Since virus-infected and tumor cells have evolved evasion mechanisms to reduce the expression of activating ligands (e.g., proteolytic shedding), the combined activity of distinct activating receptors could guarantee NK cell immunosurveillance.

Transcriptional and genetic approaches in mice as well as the identification of humans carrying germline mutations of selected TFs have helped to deconvolute the transcriptional requirements of NK cells [76,77,78]. Murine NK cells differ from other ILCs for the selective expression of the TF Eomesodermin (Eomes) [79,80,81], while other TFs originally considered NK cell-specific, among innate lymphocytes, are also required for other ILCs, such as T-bet, which controls ILC1 and NCR^+^ ILC3 [82,83,84,85,86], or Nfil3 [87,88,89,90,91,92,93,94,95,96] and Id2, which are required for the generation of the whole innate lymphoid compartment [85,97,98,99,100]. In humans, mutation of the *GATA2* gene leads to altered differentiation of NK cells and higher susceptibility of opportunistic infections and cancer [101]. As well, *T-BET* and *IRF-8* deficiency leads to defects in the differentiation and functions of human NK cells and innate-like adaptive lymphocytes [102,103,104].

Selective expression of Eomes and perforin-dependent cytotoxic functions are considered, at least in mice, two key factors to discriminate NK cells from ILC1. On the other hand, both definition of human ILC1 and identification of specific TFs have been, and still are, more problematic [105]. Several observations blur the line separating NK cells from ILC1 identities, and we will consider this issue below. Recent findings have provided evidence for the existence of ILC1 subsets expressing Eomes and/or perforin and granzymes and able to efficiently kill target cells [26]. In the intraepithelial compartment of the human and mouse intestine, a subset of ILC1 (ieILC1) expresses Eomes and can exert cytotoxic activity [28]. These cells develop in the absence of *Il15ra* in mice and accumulate during colitis in both humans and mice [28]. In the TME of human head and neck cancer, PB NK cells can differentiate towards cells that resemble ieILC1 endowed with potent in vivo antitumor activity [27]. In contrast, conversion of NK cells into ILC1-like cells has been reported as a mechanism of tumor immune evasion in the methylcholanthrene (MCA)-induced fibrosarcoma cell line MCA1956 mouse model [106]. Thus, further fate-tracing and functional studies are needed to discriminate between ILC1-like NK cells with anti- and pro- tumor potentials. Recently, our group [29] and others [34,44] have found murine Eomes^−/lo^ ILC1 able to express perforin and granzyme A, B, and C and to kill target cells. Liver granzyme A^+^ ILC1 are sensitive to Hobit deletion [30,44] and can originate from fetal ILC1 [107] or as a product of ILC3–ILC1 plasticity [29], but not from NK cells. In humans, a population of cytotoxic ILC1 was also found in the gut in conditions of IBD; this population is characterized by the expression of CD127, CD94, granulysin, and perforin and the presence of features of both CD127^+^ ILC1 and CD94^+^ NK cells [31]. In light of these phenotypic and functional similarities as well as of NK–ILC1 plasticity occurring in both physiological and pathological conditions, further work is needed to better classify the spectrum of cytotoxic and non-cytotoxic ILC1, their mechanisms of recognition, and their functions.

## 3. NK Cells and Cytotoxic Innate Lymphocytes in Intestinal Cancer

The role of NK cells in CRC was initially assessed by retrospective analysis of clinical data and by employing mouse models recapitulating distinct features of human CRC [108]. A general understanding is that both a low degree of NK cell infiltration and/or impaired NK cell functions are associated with poor overall patient survival and CRC relapse after treatment [20,109,110]. Findings obtained in mice parallel these observations, since the frequency of tumor-infiltrating NK cells progressively decreases throughout tumor progression both in the hereditary and in the sporadic mouse models of CRC [111]. In these models, in vivo administration of an antibody targeting NK1.1-expressing cells vastly exacerbates tumor formation, suggesting a role for NK cells in limiting CRC growth [111]. Since anti-NK1.1 administration also depletes ILC1, a subset of ILC3 and NKT cells, genetic approaches need to be employed to discriminate the distinct roles of murine innate and innate-like populations in CRC development. In this context, a protective role of NK cells in mouse models of colitis has been shown by using Ncr1-Cre^Eomesfl/fl^ mice, which selectively lack NK cells [112]; however, the molecular mechanisms underlying the increased inflammation in absence of NK cells are still poorly understood.

Comprehensive transcriptomic and high-dimensional flow cytometry studies have revealed a high degree of phenotypic complexity among innate lymphocytes in CRC [22,23,24,25,113,114,115,116]. Thus, together with NK cells, ieILC1-like cells represent the most abundant tumor-infiltrating innate lymphoid subset in CRC patients [114]. These cells express high levels of perforin and granzymes and are particularly enriched in mismatch repair-deficient CRC tumors [115], suggesting a potential anti-tumor role for these cells in CRC patients. Recently, a CRC tissue-specific ILC1-like population has been identified by single-cell RNA-seq analysis, characterized by the expression of *TIGIT*, *CTLA4*, and *TNFRSF4* [113]. *SLAMF1* was also found on both CRC-specific ILC1 and ILC,2 and its expression on circulating ILCs has been associated with a significantly higher survival rate in CRC patients [113]. Interestingly, circulating SLAMF1^+^ ILC2 are also increased in patients with Crohn’s Disease showing less active disease [117]. Thus, SLAMF1^+^ cells might play a protective function both during intestinal inflammation and in CRC. In the colitis-induced CRC mouse model, tumor-infiltrating ILC1 decrease over time together with their functional abilities [24,116]; however, whether these cells play a role in limiting cancer growth or progression remains to be established.

ILC1 with cytotoxic functions have been brought to the attention only recently, and many functional aspects remain to be elucidated, while more findings are available about the features of NK cells in CRC. Circulating and tumor-infiltrating NK cells in CRC patients display profound differences in terms of both receptor repertoire and effector functions, showing a drastic reduction in the expression of activating receptors, including NKG2D, NKp30, NKp46, and DNAM-1, as well as in perforin-containing lytic granules [118,119,120,121,122]. These alterations result in NK cell functional impairment in terms of IFN-γ secretion, degranulation, and cytotoxicity and are paralleled by an increased expression of the inhibitory receptors, including CD85j and NKG2A [120,122] (Figure 1A and Table 1). Moreover, NK cells from CRC patients are also characterized by an altered ADCC, showing a poor response to cetuximab treatment [123]. Different studies have shown the association between the reduced expression of NKp44 and NKp46 on circulating CD56^dim^ NK cells in CRC patients [118,124] and the presence of NCR ligands in the tumor microenvironment [124]. These findings indicate that CD56^dim^ and CD56^bright^ NK cells might differently contribute to CRC progression and suggest the involvement of additional NK cell subsets in these mechanisms [125]. To this regard, several human NK cell populations characterized by intermediate phenotypes and multiple functions have been recently identified and deserve further attention [126,127,128,129,130,131,132].

To elude adaptive immune surveillance, solid tumors, including CRC, downregulate MHC-I molecules; however, specific KIR/HLA associations and CRC susceptibility to NK cell anti-tumor response remain to be elucidated [126,136,137,138]. Regarding the action of inhibitory receptors linked with NK cell dysfunction other than KIR, in several patients affected by gastrointestinal cancers, including CRC, the expression of the immune checkpoints NKG2A and TIGIT is increased on tumor-infiltrating NK cells and is correlated with impaired NK cell cytotoxicity, advanced disease stage, and poor survival [133,134,139]. Consistently, blockade of the NKG2A or TIGIT axes promotes both NK and CD8^+^ T cell effector functions and potentiates the anti-tumor immune response [133,134]. Similarly, NK cell-mediated antitumor activity is synergistically enhanced by blocking NKG2A along with PD-1 in mice [134]. Thus, NKG2A, TIGIT, and CD96 along with the PD-1 axis represent potent candidates for CRC immunotherapy. Finally, changes in tumor cells of the ligands triggering NK cell activating receptors could compromise NK cell functionality during tumor progression. Indeed, a significant reduction of tumor NKG2D and NCR-ligands together with a low expression of MHC class I molecules correlates not only with disease stages but also with a decrease of NK cell killing susceptibility [139,140]. Altogether, these observations support the use of therapeutic strategies to target novel check-point inhibitory mechanisms able to ensure the full activity of NK cells and preserving activating ligand expression on tumor cells.

## 4. NK Cells and ILC1 in CRC Liver Metastasis

CRC patients frequently develop liver metastases that often cause death [141]. The liver is enriched with immune cells and contains a high percentage of NK cells, which represent 25–30% of the total lymphocytes in humans and 15–20% in mice [142,143,144]. Although it has long been recognized that human liver NK cells are different from their circulating counterpart [145,146,147,148], detailed phenotypic analysis of these cells have been performed only recently [149,150,151,152,153,154,155,156]. Studies based on HLA-mismatched human liver transplants have allowed us to unambiguously define the features of conventional circulating and tissue-resident NK cells [154,157]. Liver-resident NK cells are characterized by unique transcription factors and surface markers expression profiles, being defined as CD56^bright^ Eomes^hi^T-bet^lo^ and characterized for the expression of the chemokine receptor CXCR6 and the tissue residence/activation marker CD69; in contrast, circulating NK cells can be distinguished for their Eomes^lo^T-bet^hi^ profile and for the absence of CXCR6 expression [157]. RNAseq data showed a lower expression of cytotoxic molecules, such as perforin and granzyme B, by resident NK cells compared to their circulating counterparts, associated with a lower ability to kill target cells in in vitro functional assays [154]. Liver-resident NK cells represent a long-lived population which is thought to accumulate upon the recruitment to the liver of a small subset of circulating Eomes^lo^T-bet^hi^ CXCR6^+^ NK cells, via the chemokine CXCL16 expressed on the surface of sinusoidal endothelial cells [154,155]. The liver microenvironment is enriched with IL-15 and TGF-β, two cytokines able to upregulate, in vitro, Eomes expression on PB NK cells along with the expression of selected adhesion molecules involved in tissue retention, including the integrins CD103 and CD49a [158]. However, it cannot be ruled out that the liver microenvironment can also promote the in situ differentiation of circulating CD34^+^ or NK cell precursors into liver-resident NK cells. These observations support the crucial role played by the hepatic microenvironment in shaping NK cell phenotype and functions by providing a specific cytokine and chemokine milieu [159].

Since the recent identification and characterization of liver-resident NK cells, only few data have become available in regard to their role both in homeostatic and in pathological conditions [143]. A general positive correlation between the frequency of liver metastasis-infiltrating NK cells and CRC patient’s overall survival has been reported [160]. Indeed, a retrospective study on a large cohort of CRC patients showed a 5-year overall survival for those patients with a higher frequency of metastasis-infiltrating NK cells [160]. In particular, the association of neo-adjuvant chemotherapy with an EGFR inhibitor or neutralizing antibody correlates with an increase of NK cell infiltrate [160]. Accordingly, blocking the EGFR pathway has been reported to promote the tumoral production of chemokines and cytokines able to enhance the recruitment and the activation of NK cells within the tumor [161]. These findings underline the importance of the liver microenvironment in shaping the immune response toward both anti- or pro-tumor activity by promoting alterations in NK cell phenotype, functions, and frequency. A reduced density of liver-resident CD56^bright^ Eomes^hi^T-bet^lo^ NK cells in colorectal liver metastasis (CLM) has also been associated with metastasis recurrence after resection [135]. Metastasis development significantly alters the liver microenvironment, as cancer cells preferentially use glycolysis rather than oxidative phosphorylation to produce energy [162]. In particular, an increased concentration of lactate was observed in CLM as a consequence of the Warburg metabolism, and this metabolite induced a strong reduction of the intracellular pH, causing mitochondrial dysfunction that led to ROS-mediated apoptosis of liver-resident NK cells [135]. The acid tumor microenvironment strongly impacted CD56^bright^Eomes^hi^T-bet^lo^ NK cells with respect to circulating CD56^dim^Eomes^lo^T-bet^hi^, whose frequency remained unaffected [135]. The ability of a cell to control the intracellular pH is strictly dependent on mitochondrial mass and CO_2_ production as a consequence of cellular respiration; in line with this observation, liver-resident-NK cells show a lower mitochondrial mass and higher ROS production with respect to CD56^dim^ NK cells [163]. These findings highlight the importance of liver-resident NK cells in the control of tumor growth and metastasis spreading. In line with these observations, a phase I clinical study investigated the effect of in situ delivery of allogeneic NK cells combined with cetuximab and high doses of IL-2 as a possible treatment for CLM [164]. The results of this study reported the safety of allogeneic NK cell adoptive transfer and showed that better patient outcome was achieved when infused NK cells displayed higher KIR–ligand mismatches.

Evidence supporting a role for innate lymphocytes in the control of CLM has been obtained also in mouse models mainly based on intrasplenic injection of tumor cells derived from a murine primary colon carcinoma (MC38) [165]. The mouse liver contains at steady state two main NKp46^+^ populations: NK cells and ILC1 [90,166,167]. Liver NK cells are defined as CD49a^−^CD49b^+^ cells [79,168], while liver ILC1 express CD49a, TRAIL, CD69, CXCR6, and the inhibitory receptor CD200R1, but lack CD49b [43,80,169]. Moreover, liver ILC1 are very heterogeneous and include populations that differ for phenotypic and functional properties. Our group recently identified functionally distinct ILC1 populations defined by the expression of granzyme A (GzmA) and CD160 [29]. These populations display a differential ability to mediate granule-dependent killing and produce cytokines. Compared with other ILC1 populations, GzmA^+^CD160^−^ ILC1 are more cytotoxic and produce less IFN-γ when stimulated with cytokines [29].

Both NK cells and ILC1 play a role in controlling hepatic metastasis; indeed, conditional deletion of NK cells (by using Ncr1iCreEomes^fl/fl^ mice) or ILC1 (granzyme A (GzmA) *Hobit^−/−^*, *Ncr1Cre^/+^Rorα^−/−^* mice) led to an increased MC38-derived liver metastatic load, thus indicating that both populations are needed to restrict liver CRC metastasization [33,170]. However, it has been proposed that NK cells and ILC1 control different steps of this process: while ILC1 interfere with the seeding of cancer cells, NK cells prevent metastatic outgrowth and progression. A differential distribution of these populations may account for these distinct functions. Indeed, ILC1s are mainly localized in the hepatic sinusoids and outside metastatic nodules, whereas NK cells can infiltrate MC38 metastases [33]. However, in advanced metastatic disease, ILC1 are still functional and able to kill tumor cells, while NK cells lose their cytotoxic ability. The metastatic microenvironment deeply affects the liver NK cell phenotype, leading to the generation of novel NK cell subsets. Notably, NK cells appear differently re-modelled in metastasis derived from colon or lung cancer cells, suggesting that tumor-specific features play a key role in these mechanisms. In particular, MC38-induced metastases concur to generate a unique NK cell subset expressing ILC1 markers, such as Thy1, CD69, CD27, and CD49a. This subset, defined as Eomes^+^CD49a^+^CD49b^+^, is supposed to arise from NK cells in the presence of TGF-β and IL-15. Consistently, this NK cell population is enriched in genes typically associated with TGF-β and IL-15 signaling [33] and appears in metastatic livers in *Hobit^−/−^* but not in *Ncr1iCreEomes^fl/fl^* or *Ncr1iCreTgfbr2^fl/fl^* mice. These findings confirmed the relevance of the tumor microenvironment in shaping the phenotype and function of immune effector cells.

The anti-metastatic efficacy of liver NK cells is regulated by the inflammasome pathway. Mice lacking key inflammasome components, such as caspase 1 or caspase 11 (*Ice^−/−^* or *Nlrp3* (*Nlrp3^−/−^* ), have exacerbated CLM after MC38 intrasplenic injection, which supports the requirement of this pathway to suppress CRC growth in this organ [171]. In particular, IL-18 plays a critical role in inflammasome-mediated immune surveillance against metastasis. Indeed, ablation of IL-18 signaling in *IL-18^−/−^* and *Il18r1^−/−^* mice also resulted in higher tumor burden in the liver, and consistently recombinant IL-18 was able to reduce metastasis growth in *Ice^−^*^/–^ mice. The cytokine operates independently of the adaptive immune system, by promoting the anti-tumor activity of NK cells. Indeed, depletion of NK cells via anti-asialo GM1 antibodies blocks the anti-metastatic effect of IL-18, while specific loss of IL18r on NK cells recapitulates the impairment of immune surveillance found in inflammasome-deficient mice. IL-18 promotes NK cell maturation because the frequencies and numbers of fully mature NK cells (CD11b^hi^ CD27^lo^) significantly decreases in IL-18^−/−^ mice, but it also regulates FasL surface expression on NK cells, thus impacting on their cytotoxic activity [171].

The anti-metastatic role of the IL-18 pathway and NK cells in the liver is also controlled by IL-1R8, a member of the IL-1R family serving as a checkpoint for NK cell maturation and effector functions [172]. *Il1r8*^−/−^ mice exhibit protection against MC38 liver metastasis, but IL-18 genetic deficiency abrogates this effect, thus indicating a role for IL-1R8 in promoting the metastasization process via blocking inflammasome-mediated immune surveillance. In particular, adoptive transfer of *Il1r8*^−/−^ NK cells significantly reduced liver metastatic load, demonstrating a direct contribution for liver IL-1R8^+^ NK cells in these mechanisms.

The antimetastatic function of NK cells is not restricted to the liver but was also demonstrated for pulmonary CRC metastases. T-bet-deficient mice lack fully functional peripheral CD27^lo^KLRG1^+^ NK cells and are more highly susceptible to lung CRC metastases, suggesting that terminally developed NK cells are required for controlling the metastatic disease [173]. Accordingly, the adoptive transfer of wild-type CD27^lo^KLRG1^+^ NK cells succeeded in partially protecting *Tbx21^−/−^* mice from metastasis growth. Moreover, *Prf1^−/−^* but not *Ifng^−/−^* mice developed pulmonary CRC metastases in a way comparable to T-bet-deficient mice, indicating that cytotoxic activity is key for the protective function of NK cells.

Although very few and limited human and murine studies indicate a role for NK/ILC1 in the control of CRC metastasis (Figure 1B and Table 1); however, it remains poorly understood how their activity is impaired thus promoting cancer metastasization.

## 5. Concluding Remarks

It is now increasingly clear that the immune signatures associated with cancer have profound implications in the clinical development and outcome of this disease. In this context, the immune infiltrate represents a relevant prognostic factor in CRC, in which the presence of cytotoxic lymphocytes, both CD8^+^ T cell and NK cells, is associated with a better overall survival. Although NK cells and other cytotoxic innate lymphocytes are found at lower numbers than T cells in CRC and CLM specimens, their effector phenotypes should not be overlooked and rather have the potential to be employed to design novel targeted therapies and to define new biomarkers. Indeed, taking advantage of the fact that specific NK populations or ILC subsets might be found at a low level in the PB of healthy donors, these cells can represent a highly sensitive biomarker for tissue alterations/disruption if are released into the circulation, as discussed above for the expression of SLAMF1 in ILCs.

Another important aspect is that, among innate lymphocytes, ieILC1 and NK cells are the main CRC-infiltrating cells. Several findings suggest that NK cells and ieILC1 regulate different steps of CRC tumorigenesis. With the rapid advances of spatial transcriptomic approaches, it will be possible to decipher the relative positioning of these two populations in the tumor environment, as well as their activation signatures, phenotypes, and functional abilities, which will affect future choices for therapeutic strategies. This approach will also lead to the identification of novel ILC subsets with cytotoxic potential which could play a role in the immune response against CRC. Thus, understanding the complexity of cytotoxic innate lymphocytes may provide not only further prognostic tools but also wider options for the treatment of CRC and CLM and the design of specific therapies. Indeed, being localized in the tissues, these cells have the potential not only to eliminate altered cells at an early stage of transformation but also to limit tissue invasion by cancer cells. In addition, the comprehension of the complex mechanisms impacting on the anti-metastatic potential of these lymphocytes will be key to delineate strategies to contrast metastasis growth and spreading.

## Figures and Tables

**Figure 1 ijms-23-07859-f001:**
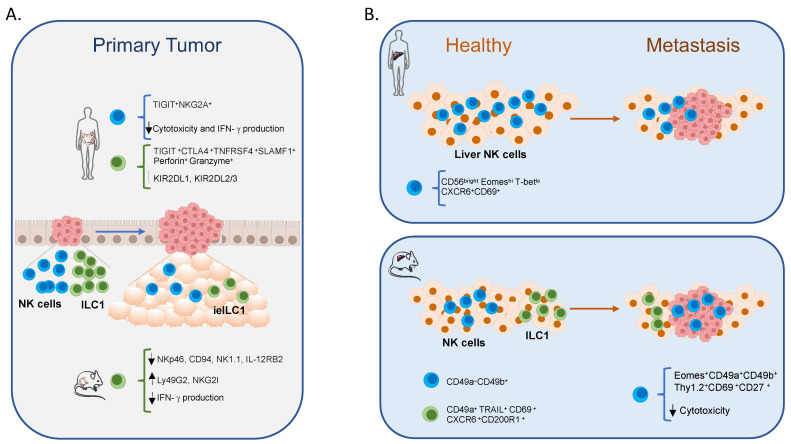
Role of NK cells and ILC1 in CRC progression and metastasis. (**A**) Both NK cells and ILC1 infiltrate CRC at an early stage to block tumor growth. However, tumor-infiltrating NK cells and ILC1 acquire a different phenotype and decrease over time together with their functional abilities. (**B**) CRC liver metastasization occurs along with the reduction of NK cells in humans and significant alterations of phenotypic and functional properties of NK cells in mice. Arrow indicates a decrease or increase.

**Table 1 ijms-23-07859-t001:** Contribution of NK/ILC1 in CRC progression and metastasis. Grey: Decrease; Bold: Increase; -: Not determined.

	PRIMARY CRC	LIVER CRC METASTASIS
Phenotype	Function	Ref.	Phenotype	Function	Ref.
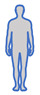	NK	CD16, NKG2D, NKp30, NKp46, CD161, DNAM-1, CD158b, CD158a/h **NKp44** **TIGIT^+^ NKG2A^+^**	-Frequency at late stage-Degranulation -IFN-γ, TNF production-Low infiltrate and functional impairment associated with poor overall survival and relapse	[122][133][134][20][109][110]	CD56^bright^CXCR6^+^TRAIL^+^CD69^+^NKG2D, NKG2C,NKp46, NKp44,PERFORIN^+^GRANZYME^+^	-Apoptosis-induced depletion-A reduced density associated with metastasis recurrence.	[135]
ILC1	TIGIT^+^ CTLA4^+^ TNFRSF4^+^SLAMF1^+^PERFORIN^+^GRANZYME^+^**KIR2DL1, KIR2DL2/3**	-Frequency at late stage	[113][116]			
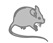	NK	**TIGIT, Ly49a, CD96,****LAG3, TIM3**NKG2D, DNAM-1	-Frequency at late stage-Degranulation-IFN-γ, TNF production	[111][133]	**Eomes^+^CD49a^+^CD49b^+^** **Thy1.2^+^CD69^+^CD27 ^+^**	-Infiltrate the metastatic nodules -Restrain metastasis growth-Cytotoxicity at late stage	[33]
ILC1	NKp46, CD94, NK1.1, IL-12RB2 **Ly49G2, NKG2I**	-Frequency at late stage-IFN-γ production	[116]	-	-Outside the metastatic nodules-Control metastasis seeding-Cytotoxicity at late stage	[33]

## Data Availability

Not applicable.

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
