# Peer review of "NK Cells and Other Cytotoxic Innate Lymphocytes in Colorectal Cancer Progression and Metastasis"

_ijms, 2022, doi:10.3390/ijms23147859_

Round 1

Reviewer 1 Report

The manuscript entitled "NK cells and other cytotoxic innate lymphocytes in colorectal cancer progression and metastasis" presents an extensive review of the immune mechanisms deployed by innate cytotoxic lymphocytes to fight tumors (specifically Colorectal cancer).
The review presents a clear classification of cytotoxic lymphocytes along with aspects of their regulation, the infiltration capabilities of the different types of cells, their distribution in tumor-affected tissues, and their behavior depending on the changes in the microenvironment (such as those induced by metastases). They also highlight the potential prognostic/therapeutic of the cells mentioned above as we progress on deciphering their regulation mechanisms and full capabilities.
In my opinion, this is a useful review and should be published in this Journal.

Author Response

We thank the reviewer for the overall positive comment.

We have fixed the typos present in the manuscript. 

Reviewer 2 Report

The presented manuscript is concerning the features of NK cells and the expanding spectrum of innate lymphocytes with cytotoxic functions, the potential role of innate lymphocytes during progression of intestinal cancer and its metastasis, as well as recent advances on the molecular mechanisms underlying functional regulation of cytotoxic innate lymphocytes in colorectal cancer.

Paper briefly summarizes the aim of the study and is divided into individual sections in which the authors accurately explain the assumed goals. In terms of content, the information was presented fairly and accurately.

The work is an extremely valuable, however a few minor changes could enrich the manuscript:

1. Described figures (1A and 1B) are not included in the main text. This should be completed.

2. The main text, despite many references (173 items), has only 4388 words. I propose to expand the mentioned section on NK cells functions (page 5).

3. I propose to add a table summarizing the achievements in the CRC so far. A compilation of the participation of the described cells, taking into account the division into primary cancer and metastases, would be very interesting.

4. The work should be edited by an English native speaker.

Author Response

The work is an extremely valuable, however a few minor changes could enrich the manuscript:

We thank the reviewer for the comments which helped to improve our manuscript. 

  1. Described figures (1A and 1B) are not included in the main text. This should be completed.

We have now added this information within the main text.

  1. The main text, despite many references (173 items), has only 4388 words. I propose to expand the mentioned section on NK cells functions (page 5).

We thank the reviewer for this comment. We have now expanded this part. 

  1. I propose to add a table summarizing the achievements in the CRC so far. A compilation of the participation of the described cells, taking into account the division into primary cancer and metastases, would be very interesting.

According to the reviewer suggestion, we have now added a table summarizing the main findings subdivided by species and cell type. 

  1. The work should be edited by an English native speaker.

We thank the reviewer for this comment. The work has been edited and the typos fixed.